# Effects of Gamma Radiation Doses on the AC Electrical Properties of Epoxy Reinforced with Nano-Silica Composites

Batool A. Abusaleh [1,*], Ziad M. Elimat [2], Ruba I. Alzubi [3] and Hassan K. Juwhari [4]

1    Department of Applied Science, Faculty of Ma'an College, Al-Balqa' Applied University, Al-Salt 19117, Jordan
2    Department of scientific Basic Sciences, Faculty of Engineering Technology, Al-Balqa' Applied University, Al-Salt 19117, Jordan; zaid-elimat@bau.edu.jo
3    Jordan Atomic Energy Commission, Amman 11934, Jordan; ruba.alzubi@jaec.gov.jo
4    Physics Department, School of Science, University of Jordan, Amman 11942, Jordan; h.juwhari@ju.edu.jo
*    Correspondence: batool.abusaleh@bau.edu.jo

**Abstract:** This study reports the effects of gamma radiation on the AC electrical properties of epoxy nano-silica composite sheets with an average thickness of 3 mm. Epoxy reinforced with different nano-silica concentrations of 0, 5, 10, and 15 wt.%. The epoxy nano-silica composites were exposed to different gamma radiation dosages of 1, 3, and 5 kGy. The data were analyzed before and after gamma irradiation and the results showed that the AC electrical properties of the gamma-irradiated epoxy nano-silica composites varied from those of the non-irradiated samples. We found that there were significant changes in the impedance, dielectric constant, dielectric loss, and AC electrical conductivity values of the epoxy nano-silica composites after irradiation, and the AC electrical conductivity and dielectric constant values of the nano-silica composites were enhanced by exposing the samples to gamma radiation because more free electrons were released inside the epoxy nano-silica composites. The AC electrical properties, such as impedance, dielectric constant, dielectric loss, and electrical conductivity, of the epoxy nano-silica composites were studied and discussed before and after gamma radiation with different dosages. We found that these AC electrical properties marginally increased with increasing doses of gamma irradiation.

**Keywords:** epoxy; nano-silica; composites; gamma radiation; dielectric; conductivity

## 1. Introduction

Nanotechnology has been around for centuries, but it has only been in the last 40 years that we have truly begun to understand the potential of this technology. Nanotechnology is the study and manipulation of matter at the atomic and molecular level. This allows us to create materials and devices with unique properties that cannot be achieved with traditional materials.

One of the most promising areas of nanotechnology is nanocomposites. Nanocomposites are materials that are made up of a matrix material, such as a polymer or metal, and nano-sized filler material. The filler material can be made of a variety of materials, including metals, ceramics, and polymers. The addition of the nano-sized filler material to the matrix material can significantly improve the properties of the material, such as its strength, stiffness, toughness, and electrical conductivity.

Nanocomposites have the potential to revolutionize many industries. Some of the potential applications of nanocomposites include:

-    Healthcare: Nanocomposites can be used to create new medical devices, such as drug delivery systems and tissue engineering scaffolds. Energy: Nanocomposites can be used to create new energy storage devices, such as batteries and fuel cells.
-    Transportation: Nanocomposites can be used to create lighter, stronger, and more fuel-efficient vehicles.

- Construction: Nanocomposites can be used to create stronger, more durable, and more energy-efficient buildings.
- Consumer goods: Nanocomposites can be used to create new and improved consumer goods, such as clothing, electronics, and food packaging.

The potential applications of nanocomposites are vast and still being explored. Nanocomposites have the potential to revolutionize many industries and improve our lives in many ways [1–3].

The effects of gamma radiation on the physical properties of polymer nanocomposites have been extensively studied. In general, radiation-induced secession or cross-linking alters the degree of aggregation of filler particles and creates holes or cavities in irradiated composites [4,5].

Furthermore, radiation may cause changes in the crystallization, chemical structure, thermal stability, mechanical properties, and morphology of polymer composites. Thus, irradiation is an important factor to consider when using polymer composites in various applications. Exposure of nanomaterials to high-energy ionizing radiation, such as gamma radiation, can alter and improve the physical properties, inert structure, and behavior parameters of the polymeric nanocomposite materials [6].

This review article Parida, S. K. et al. rovides an extensive overview of polymer nanocomposites, including their structure, properties, manufacturing techniques, and potential applications. Polymer nanocomposites are materials that are made by dispersing a small number of nanosized particles in a polymer matrix. The nanoparticles can be made of a variety of materials, including metals, ceramics, and polymers. The addition of nanoparticles to a polymer matrix can significantly improve the properties of the material, such as its strength, stiffness, toughness, and electrical conductivity. The exceptional properties of polymer nanocomposites are due to the large surface area of the nanoparticles. The nanoparticles have a very high surface area, which means that they can interact with a large number of polymer chains. This interaction between the nanoparticles and the polymer chains creates a strong bond that makes the material much stronger and tougher than the original polymer [7].

The present study aims to analyze the effect of gamma radiation doses on the AC electrical properties of epoxy resin filled with different concentrations of silica nanoparticles. Several scientific research papers in the literature reported the effect of gamma radiation on the physical properties of silica and/or epoxy polymer nanocomposites; following are some examples: Babu et al. have stated that when exposed to epoxy nanocomposites to gamma radiation; It increases the space charge density and real permittivity with an increase in dosage of gamma irradiation due to the formation of free radicals in the bulk of the nanocomposites material [8]. Abd Allah et al. studied the effect of gamma radiation on the dielectric properties of epoxy filled with silicon dioxide and titanium dioxide, and they concluded that the values of the dielectric parameters of the tested samples increased with an increase in the gamma irradiation dosage [9]. Karpagam et al. studied the influence of gamma irradiation on the electrical and mechanical properties of the epoxy clay nanocomposites and reported that gamma irradiation of the epoxy nanocomposite material resulted in an increase in permittivity [10]. Researchers Anwaret et al. studied the effect of gamma irradiation on the optical and electrical properties of Fiberglass/epoxy and Kevlar fiber/epoxy composites and found that the electric conductivity and dielectric constant for those samples increased with increasing gamma radiation doses [6]. Liet al. studied the effect of gamma irradiation on the electrical properties of an epoxy encapsulate and reported that both the relative permittivity and dissipation factor of the epoxies increased with the irradiation dose [6]. Ismaiilova et al. studied the effect of gamma-Irradiated nanocomposites based on ultrahigh-molecular-weight polyethylene filled with alpha-$SiO_2$ and found that the temperature dependence of the electrical conductivity of all samples after exposure to ionizing radiation undergoes serious changes [11]. Craciun et al. studied the radiation behavior of nanocomposite epoxy material and reported that the electrical behavior is controlled by the irradiation dose [12]. Patel et al. studied the

gamma radiation-induced effects on silica and the interfacial interactions between silica and polymer in filled polysiloxane rubber, and they reported that the silica surface is sensitive to gamma radiation [13].

In the literature, there are no published papers or reported data concerning the effects of gamma irradiation on the electrical properties of epoxy-filled nano-silica composites, which encourages us to study the present research. Previously, we have reported on the optoelectrical properties of epoxy/silica nanocomposites [14]. The present study deals with the effect of gamma irradiation on the AC electrical properties of epoxy nano-silica composites.

There are a number of different methods that can be used to manufacture polymer nanocomposites. The choice of manufacturing method depends on the type of polymer and the nanoparticles that are being used. The most common methods are [15]:

1. In situ polymerization: This method involves polymerizing the polymer in the presence of the nanoparticles.
2. Melt blending: This method involves melting the polymer, then adding the nanoparticles to the molten polymer.
3. Solution blending: This method involves dissolving the polymer in a solvent and then adding the nanoparticles to the solution. This method was used to prepare epoxy nano-silica composites with different silica concentrations = 0, 5, 10, and 15 wt.%, respectively.

The use of gamma radiation to alter the material properties of epoxy composites opens up a range of new possibilities. This technique has the potential to create advanced materials with improved electrical insulation, and improved electrical conductivity. The applications of these materials could be used in the range from high-voltage cabling and transformers to EMI shielding and heat sinks. By making tweaks to the properties of epoxy composites with gamma radiation, it could be possible to substitute the use of metals in heat transfer and the mitigation of electromagnetic interference. The potential applications for this technology are vast and could have far-reaching implications for materials development [1,15,16].

## 2. Experimental

### 2.1. Samples Preparation

The epoxy resin was supplied by Ciba-Geigy. The resin hardener was 4,4′-diaminodiphenylsulphone, received from Aldrich. The silica component was obtained from Clariant as a 30% silicamicro-dispersion in isopropanol, sold under the trade name of High-link OG 502-30. The nanocomposites contain silica primary nanoparticles with a diameter of 10–15 nm. Measurements were made on disk-shaped specimens. Specimens 3 mm in thickness were cut from the prepared nanocomposite sheets. More details about the nanocomposite preparation were listed in our previously published paper [14].

To study the electrical properties of epoxy nano-silica composite, impedance (Z) and phase shift angle (φ) measurements were performed using a Hewlett Packard (HP) 4192A Impedance Analyzer. The HP 4192A Impedance Analyzer measures impedance and phase angle as a function of applied frequency.

### 2.2. Gamma Radiation (γ Radiation)

The gamma radiation was incorporated to investigate the influence of the AC electrical properties of the epoxy nano-silica composites. Samples were irradiated with different gamma radiation dosages = 1, 3, and 5 kGy by Co-60 gamma-ray source with an average energy of 1.25 MeV with a dose rate of 205.965 Gy/hr for a sample capacity of 4400 mL at Jordan Atomic Energy Commission.

### 2.3. AC-Electrical Parameters Calculations

Electrical properties and parameters of the composites were calculated using standard formulas as listed as the followings: The complex impedance $(Z^*)$ of the sample with the real $(Z_r)$ and imaginary $(Z_i)$ components can be calculated by [16]:

$$Z^* = Z_r + iZ_i \tag{1}$$

$$Z_r = Z\cos \varphi \text{ and } Z_i = Z\sin \varphi \tag{2}$$

where $(Z)$ is the impedance magnitude and $(\varphi)$ is the phase angle, both measured by HP impedance analyzer. The AC-conductivity of the sample is calculated from the following equation:

$$\sigma_{ac} = 2\pi f \epsilon_o \varepsilon_i \tag{3}$$

where $(f)$ is the applied frequency and $(\epsilon_o)$ is the permittivity of free space. Additionally, the $(\varepsilon_i)$ dielectric loss is given by:

$$\varepsilon_i = \frac{Z_r}{2\pi f C_o Z^2} \tag{4}$$

and $(C_o)$ is the capacitance of two plates of the sample given by:

$$C_o = \epsilon_o \frac{A}{d} \tag{5}$$

where, $(A)$ is the disk area and $(d)$ is the distance between the two plates.

## 3. Results and Discussion

The impedance of a material is a measure of its opposition to the flow of an electric current. The higher the impedance, the more difficult it is for an electric current to flow through the material [17].

The decrease in impedance with increasing irradiation dosages is due to the formation of free radicals in the polymer matrix. Free radicals are atoms or molecules that have unpaired electrons. These unpaired electrons can act as charge carriers, which can increase the conductivity of the material. The increase in conductivity leads to a decrease in the impedance [18].

The effect of gamma radiation on the impedance of epoxy composites is more pronounced at higher nano-silica concentrations. This is because the nano-silica particles can act as traps for free radicals. The trapping of free radicals reduces the number of free radicals that are available to act as charge carriers, which leads to a smaller decrease in the impedance [16].

Figure 1a–d shows that the impedance of epoxy composites filled with nano-silica decreases with increasing irradiation dosages. This decrease in impedance is due to the formation of free radicals in the polymer matrix. The effect of gamma radiation on the impedance of epoxy composites is more pronounced at higher nano-silica concentrations. The impedance of the epoxy/nano-silica composites is approximately between 1100 kΩ for pure epoxy and 900 kΩ for the highest concentration of nano-silica (15%) shown in Figure 1a before irradiation. For exposing the nanocomposites to 1 kGy gamma irradiation, the impedance decreases drastically, approximately between 950 kΩ for pure epoxy and 700 kΩ for the highest concentration of silica, as shown in Figure 1b. Figure 1c also shows that the impedance continues to decrease steadily with increasing doses of γ-radiation to 3 kGy, reaching a value of approximately between 730 kΩ for pure epoxy and 590 kΩ the highest concentrations. In the same situation Figure 1d shows the impedance decreased with the increase in radiation to 5 kGy, as it was observed that it decreased to 540 for pure epoxy and to 410 kΩ for highest concentration. The increase in the gamma irradiation

dosage may increase the number of free radicals that react to decrease the impedance of the tested samples [9,16,19].

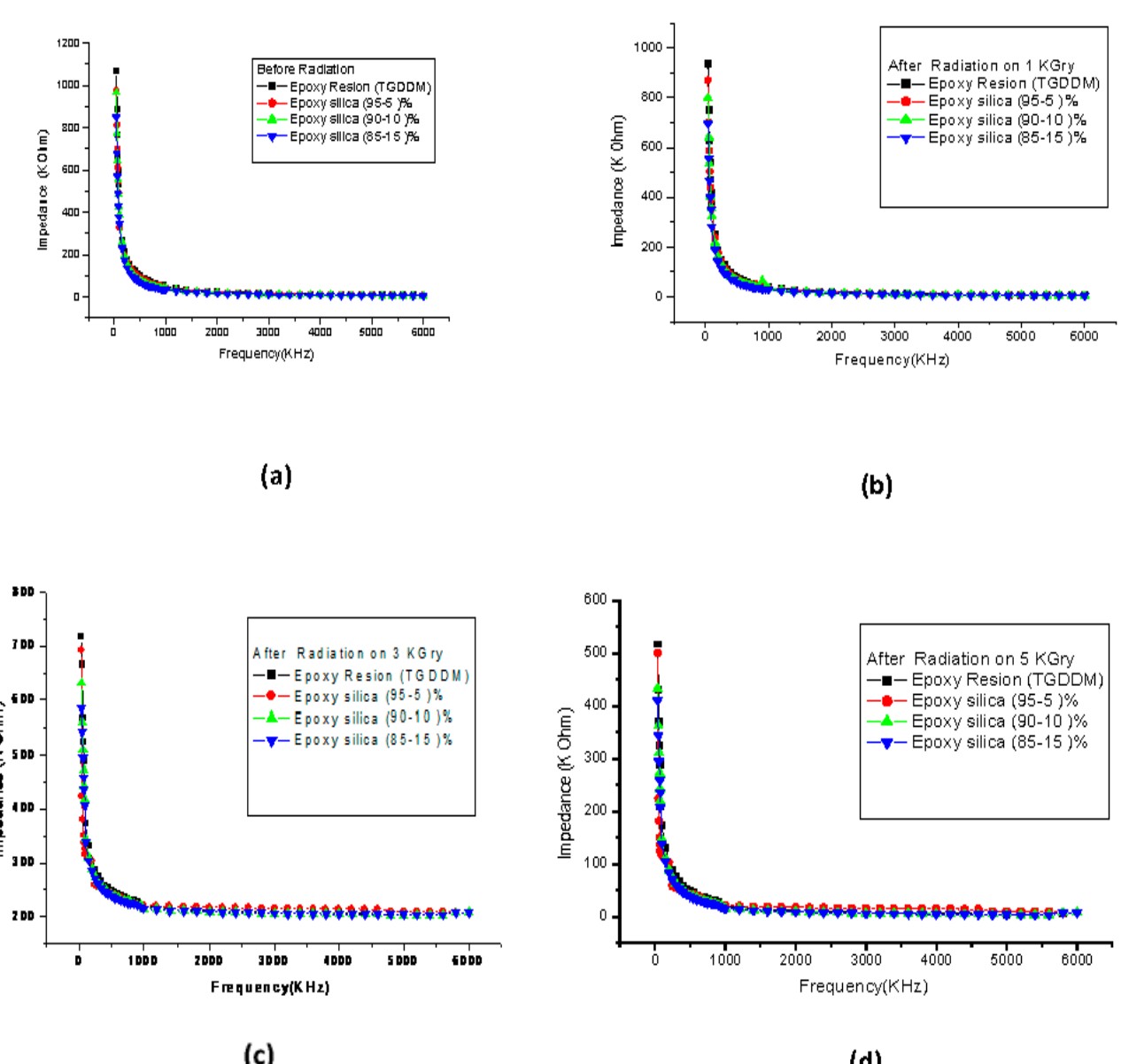

**Figure 1.** (**a**–**d**) The dependence of impedance on versus frequency of epoxy filled with different nano-silica concentrations composites at room temperature before and after gamma radiation.

Figure 2a–d shows the dielectric constant frequency dependence of epoxy filled with different nano-silica concentrations composites at room temperature before and after gamma radiation. The dielectric constant of a material is a measure of its ability to store electric charge. The higher the dielectric constant, the more easily a material can store electric charge [20].

The decrease in dielectric constant with increasing gamma radiation dosages is due to the formation of free radicals in the polymer matrix. Free radicals are atoms or molecules that have unpaired electrons. These unpaired electrons can interact with the electric field of the material, which can reduce the ability of the material to store electric charge. The decrease in the ability to store electric charge leads to a decrease in the dielectric constant [21].

The effect of gamma radiation on the dielectric constant of epoxy composites is more pronounced at higher nano-silica concentrations. This is because the nano-silica particles can act as traps for free radicals. The trapping of free radicals reduces the number of free radicals that are able to interact with the electric field of the material, which leads to a smaller decrease in the dielectric constant [21].

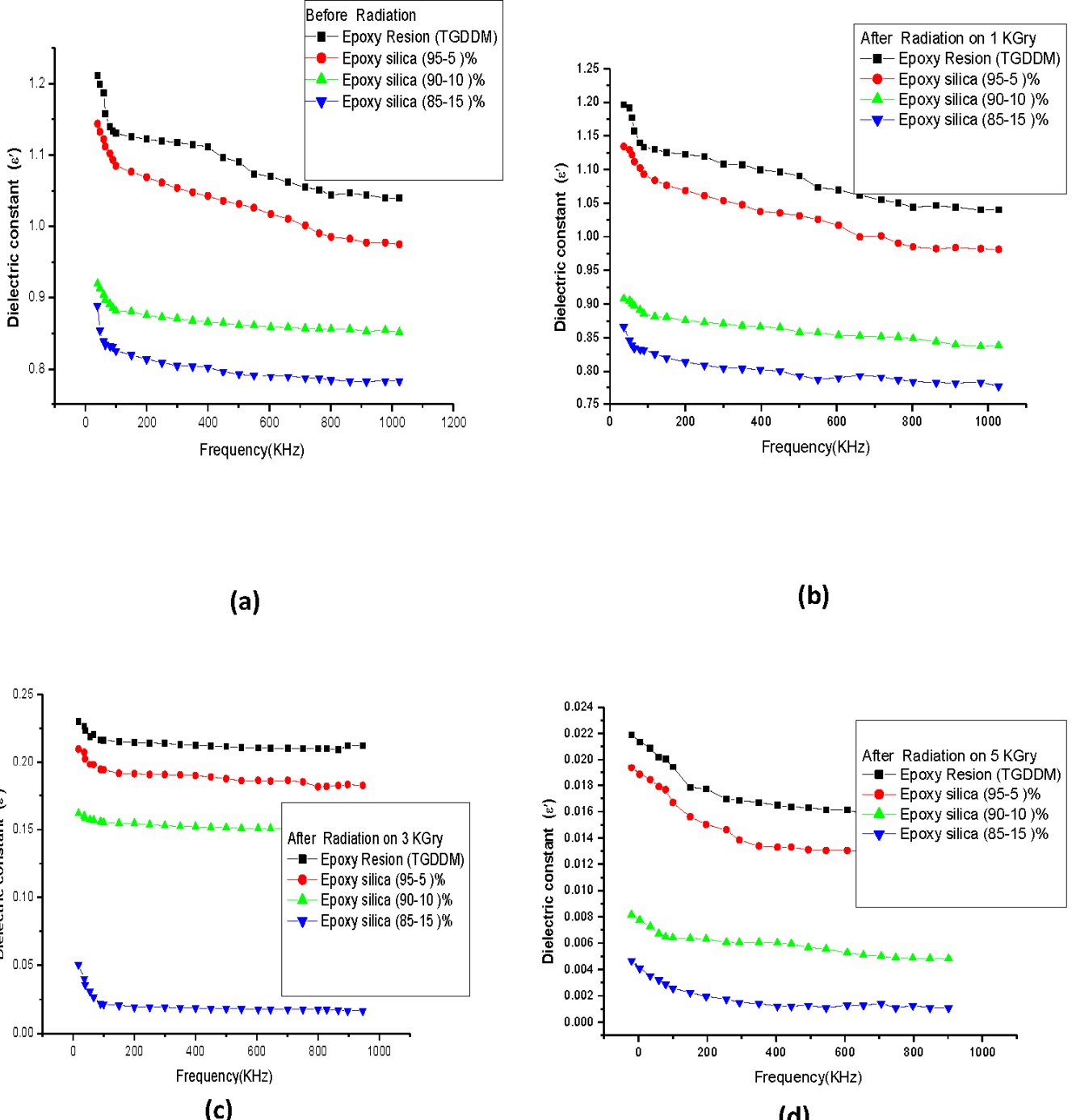

**Figure 2.** (**a**–**d**) The dependence of dielectric constant versus frequency of epoxy filled with different nano-silica concentrations composites at room temperature before and after gamma radiation.

The results of this study show that the dielectric constant values were decreasing with increasing γ-radiation dosages. The dielectric constant of the epoxy/nano-silica composites is approximately between 1.25 for pure epoxy and 0.9 for highest concentration of nano-

silica (15%), shown in Figure 2a before γ-radiation. For exposing the nanocomposites to 1 kGy gamma irradiation, the dielectric constant decreased drastically approximately between 1.20 for pure epoxy and 0.87 for the highest concentration of silica, as shown in Figure 2b. Figure 2c also shows that the dielectric constant continues to decrease steadily with increasing doses of γ-radiation to 3 kGy, reaching a value of approximately between 0.23 for pure epoxy and 0.05 for the highest concentration. In the same situation, Figure 2d shows the dielectric constant decreased with the increase in radiation to 5 kGy, as it was observed that it decreased to 0.021 for pure epoxy and to 0.006 for the highest concentration. The increase in dielectric constant with an increase in radiation dosage may be attributed to the radiation creates free radicals molecules that are induced through different mechanisms that occur inside the material, such as chain scission and cross-linking; as a result, the dipole orientation will be more and the dielectric constant should be increased [8,9,21].

Figure 3a–d shows the dielectric loss frequency dependence of epoxy filled with different nano-silica concentrations composites at room temperature before and after gamma radiation. This figure shows that the dielectric loss values were decreasing with increasing γ-radiation dosages. The dielectric loss of the epoxy/nano-silica composites is approximately between 0.00057 for pure epoxy and 0.0005 for the highest concentration of nano-silica (15%) shown in Figure 3a before γ-radiation. For exposing the nanocomposites to 1 kGy gamma irradiation, the dielectric loss decreases drastically, approximately between 0.00053 for pure epoxy and 0.00044 for the highest concentration of silica, as shown in Figure 3b. Figure 3c also shows that the dielectric loss continues to decrease steadily with increasing doses of γ-radiation to 3 kGy, reaching a value of approximately between 0.00115 for pure epoxy and 0.0006 for the highest concentration. In the same situation, Figure 3d shows the dielectric loss decreased with the increase in radiation to 5 kGy, as it was observed that it decreased to 0.00133 for pure epoxy and to 0.00100 for the highest concentration. The increase in the dielectric loss value due to the gamma irradiation is likely due to the increase in the number of dipoles created in the composites. This is because gamma radiation causes an increase in free electrons and electron-hole pairs, leading to more charges which then form dipoles. The more dipoles that are present in a material, the more energy will be lost when an electric field is applied.

The increase in dielectric loss can have a number of effects on the material. For example, it can reduce the material's ability to store energy, and it can also increase the material's susceptibility to electrical breakdown. The effects of dielectric loss will depend on the specific material and the conditions under which it is operating [9,16–18,22].

Figure 4a–d shows the AC-conductivity frequency dependence of epoxy filled with different nano-silica concentrations composites at room temperature before and after gamma radiation. This figure shows that the AC-conductivity values were increasing with increasing γ-radiation dosages. The AC-conductivity of the epoxy/nano-silica composites is approximately between $0.0047 \times 10^{-8}$ $(\Omega\cdot m)^{-1}$ for pure epoxy and $0.0067 \times 10^{-8}$ $(\Omega\cdot m)^{-1}$ for the highest concentration of nano-silica (15%) shows in Figure 4a before γ-radiation, by exposing the nanocomposites to 1 kGy gamma irradiation, the AC-conductivity increase drastically approximately between $0.0123 \times 10^{-8}$ $(\Omega\cdot m)^{-1}$ for pure epoxy and $0.016 \times 10^{-8}$ $(\Omega\cdot m)^{-1}$ for the highest concentration of silica as shown in Figure 4b. Figure 4c also shows that the AC-conductivity continues to increase steadily with increasing doses of γ-radiation to 3 kGy, reaching a value of approximately between $0.017 \times 10^{-8}$ $(\Omega\cdot m)^{-1}$ for pure epoxy and $0.035 \times 10^{-8}$ $(\Omega\cdot m)^{-1}$ for highest concentration. In the same situation, Figure 4d shows the AC-conductivity increased with the increase in radiation to 5 kGy, as it was observed that it increased to $0.0355 \times 10^{-8}$ $(\Omega\cdot m)^{-1}$ for pure epoxy and to $0.075 \times 10^{-8}$ $(\Omega\cdot m)^{-1}$ for highest concentration. Gamma radiation is electromagnetic radiation that has a high energy level, which makes it able to ionize atoms by removing electrons, thus creating positively and negatively charged ions, so these increase the degree of dipole orientation, then increase the AC electrical conductivity. Additionally, when the gamma dosage is increased, it can lead to scission of branched chains, which increases the mobility of the free charge carriers and thus increases electrical conductivity [23–25].

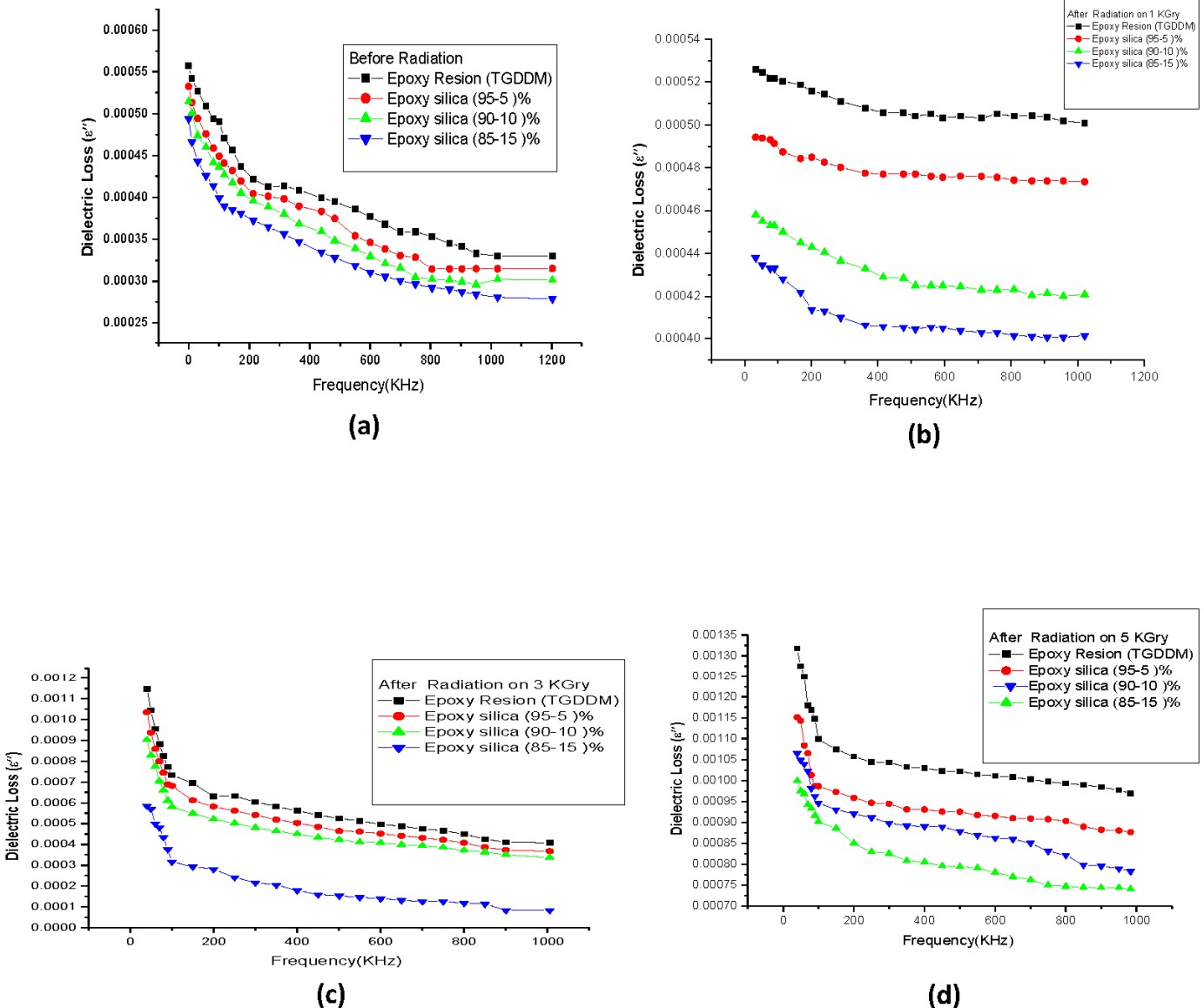

**Figure 3.** (**a–d**): The dependence of dielectric loss versus frequency of epoxy filled with different nano-silica concentrations composites at room temperature before and after gamma radiation.

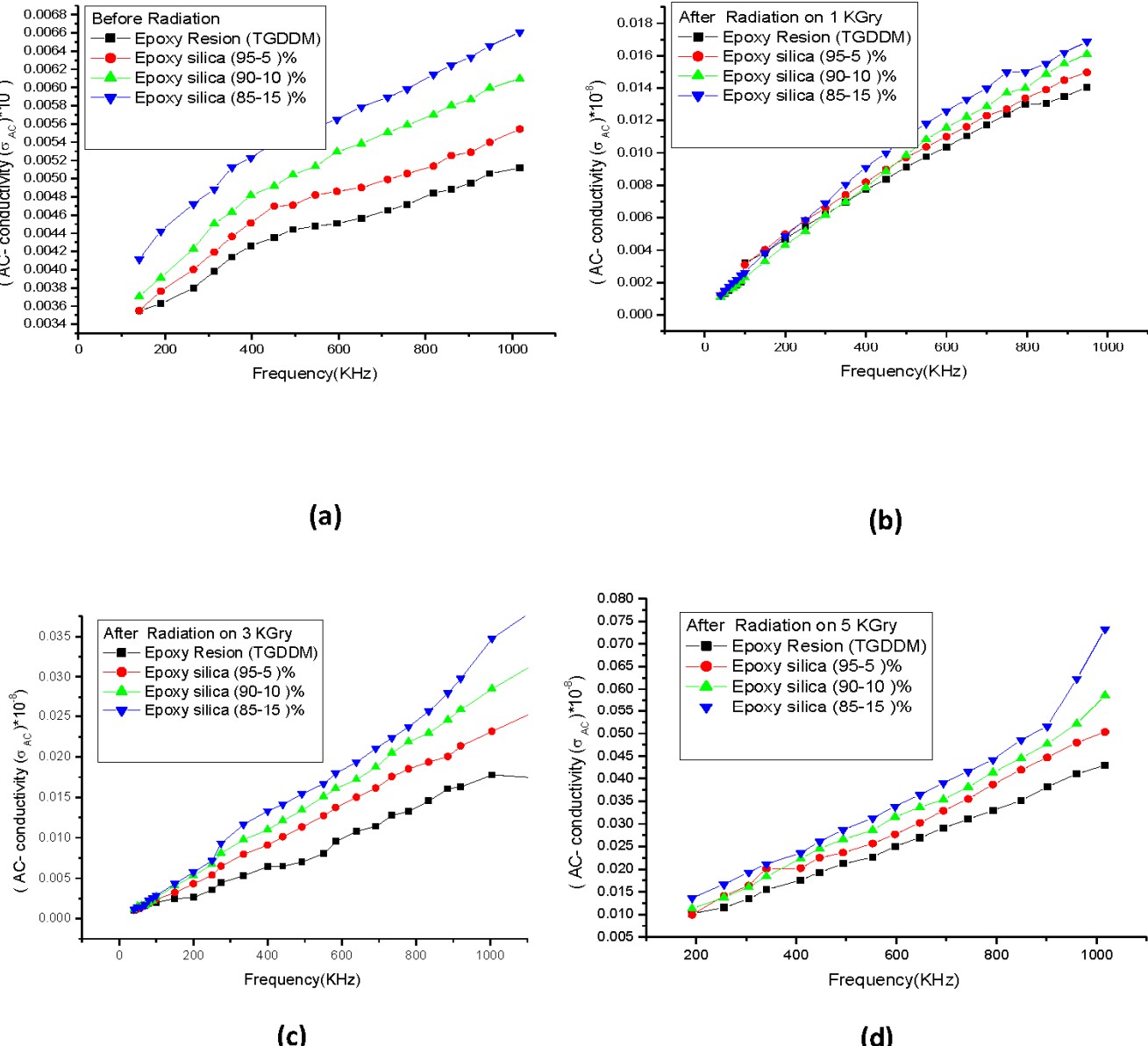

**Figure 4.** (**a–d**) The dependence of AC-conductivity versus frequency of epoxy filled with different nano-silica concentrations composites at room temperature before and after gamma radiation.

The Jonscher's Universal Law Model utilizes the length distribution of conduction paths that are accessible for electric charge flow to explain the universal power-law dispersive AC conductivity seen in polymer networks and disordered matter. Values close to 1 are physiologically acceptable within this model and are in agreement with experimental results [26]. It also predicts saturation in the high-frequency region of AC conductivity, which has been seen in conducting polymer composites, taking the form of a power-law model [27]:

$$\sigma_{ac} = \sigma_0 + Af^n$$

The coefficient values A and n are provided in Tables 1–4 and $\sigma_0$ represents the plateau in $\sigma_0$, which is usually associated with the DC conductivity of the material. The plots in Figure 5a–d demonstrate that, at high irradiation, the conductivity of the material increases as a power of frequency with a power exponent close to one.

It can be concluded that, under the condition of $\sigma_o$ being much less than $\sigma$, the equation given above can be simplified to:

$$\sigma_{ac} = A\ f^n$$

It was observed that the value of the power exponent (n) increases when irradiation increased for all composites. nano-silica composites that maintain semiconducting properties may be attributed to the thermal activation which causes from irradiation of their charge carriers [26].

**Table 1.** The estimated A and n coefficients for epoxy nano-silica composite before gamma radiation.

| Concentration of Nano-Silica in Composite | n | A $(\Omega \cdot m)^{-1}$ $(Hz)^{-n}$ |
|---|---|---|
| 0% | 0.10 | $8.9 \times 10^{-7}$ |
| 5% | 0.102 | $1.09 \times 10^{-6}$ |
| 10% | 0.103 | $1.11 \times 10^{-6}$ |
| 15% | 0.107 | $1.22 \times 10^{-6}$ |

**Table 2.** The estimated A and n coefficients for epoxy nano-silica composite on 1 kGy gamma irradiation.

| Concentration of Nano-Silica in Composite | n | A $(\Omega \cdot m)^{-1}$ $(Hz)^{-n}$ |
|---|---|---|
| 0% | 0.19 | $2.30 \times 10^{-6}$ |
| 5% | 0.214 | $2.46 \times 10^{-6}$ |
| 10% | 0.24 | $2.59 \times 10^{-6}$ |
| 15% | 0.25 | $2.62 \times 10^{-6}$ |

**Table 3.** The estimated A and n coefficients for epoxy nano-silica composite on 3 kGy gamma irradiation.

| Concentration of Nano-Silica in Composite | n | A $(\Omega \cdot m)^{-1}$ $(Hz)^{-n}$ |
|---|---|---|
| 0% | 0.40 | $4.32 \times 10^{-6}$ |
| 5% | 0.72 | $4.65 \times 10^{-6}$ |
| 10% | 0.75 | $5.4 \times 10^{-6}$ |
| 15% | 0.76 | $2.79 \times 10^{-5}$ |

**Table 4.** The estimated A and n coefficients for epoxy nano-silica composite on 5 kGy gamma irradiation.

| Concentration of Nano-Silica in Composite | n | A $(\Omega \cdot m)^{-1}$ $(Hz)^{-n}$ |
|---|---|---|
| 0% | 0.91 | $5.98 \times 10^{-5}$ |
| 5% | 0.93 | $6.55 \times 10^{-5}$ |
| 10% | 0.94 | $7.09 \times 10^{-5}$ |
| 15% | 0.95 | $7.95 \times 10^{-5}$ |

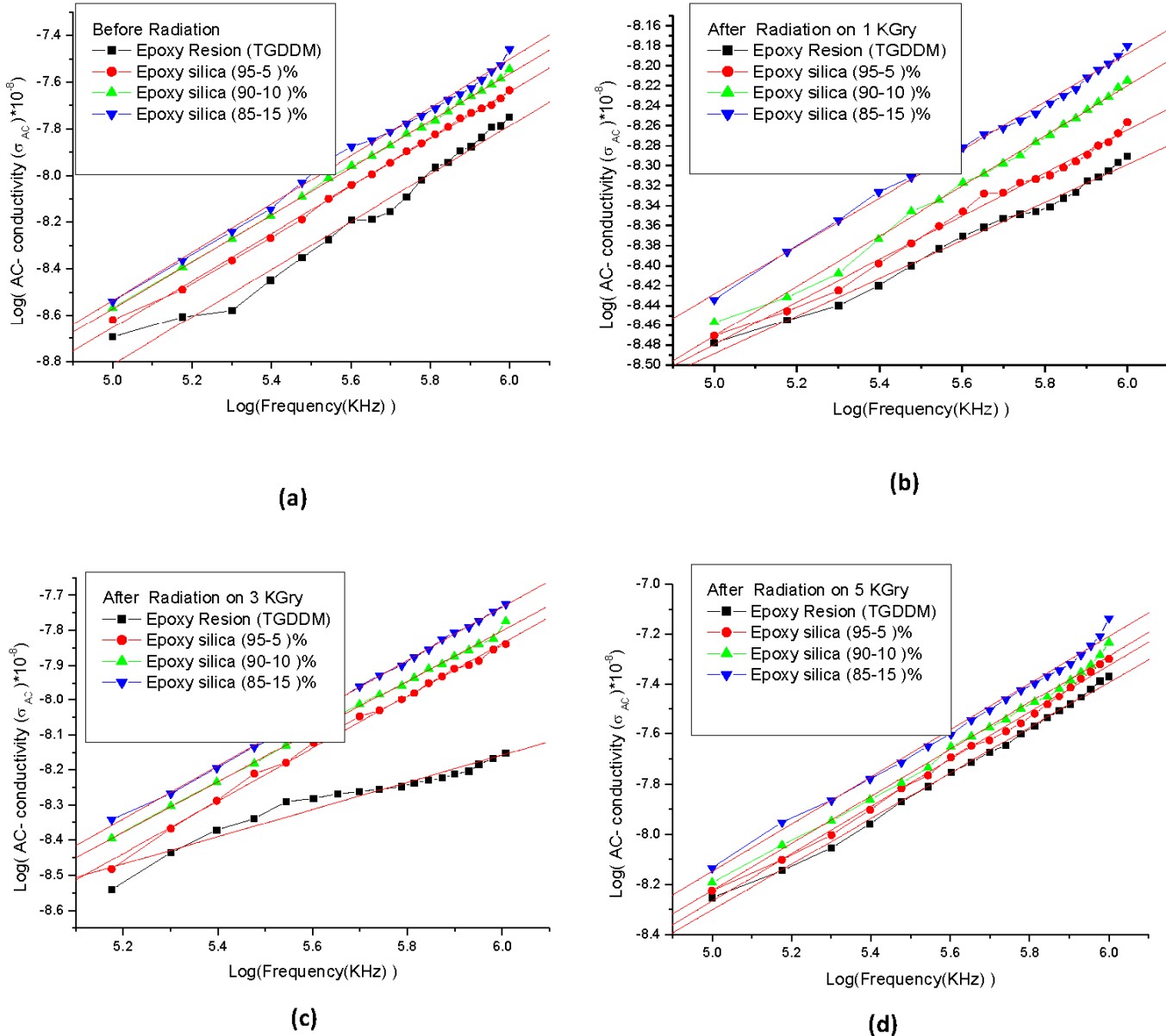

**Figure 5.** (**a**–**d**): log (AC-Conductivity) versus log (frequency) of epoxy filled with different nano-silica concentrations composites at room temperature before and after gamma radiation.

## 4. Conclusions

Epoxy nano-silica composites were irradiated with dosages of 1, 3, and 5 kGy by a Co-60 gamma-ray source with a dose rate of 205.965 Gy/h for a sample capacity of 4400 mL. The ac electrical properties of the composites were measured before and after irradiation. The results showed that the epoxy nano-silica composites were affected by the dosage of gamma radiation. The impedance values decreased for the gamma-irradiated composites. We found that gamma radiation can be used as a curing technique to decrease the electrical impedance of samples. The dielectric constant and dielectric loss of the composites also decreased after irradiation.

The gamma radiation process is a non-contact, non-thermal process that can be used to modify the properties of materials. Gamma radiation can be used to cure polymers, create cross-links in polymers, and induce an increase in ac electrical conductivity of the composites.

The gamma radiation process is a safe and environmentally friendly process that can be used to produce high-quality materials.

**Author Contributions:** B.A.A. conceived the study, designed the experiment, carried out the experiment, collected the data, analyzed the data, and wrote the initial draft of the manuscript. Z.M.E. provided support and guidance throughout the project, including help with the design of the experiment, the interpretation of the data, and the writing of the manuscript. R.I.A. irradiated the samples in Jordan Atomic Energy Commission. H.K.J. helped supervise the project and provided feedback on the manuscript. All authors have read and agreed to the published version of the manuscript.

**Funding:** This research received no external funding.

**Data Availability Statement:** All data and results were obtained through the devices in the physics laboratory at the University of Jordan and were mentioned in the research.

**Conflicts of Interest:** The authors declare no conflict of interest.

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
