# Peer review of "Effects of Gamma Radiation Doses on the AC Electrical Properties of Epoxy Reinforced with Nano-Silica Composites"

_jcs, doi:10.3390/jcs7060254_

Round 1

Reviewer 1 Report

This study reports the effects of gamma radiation on the ACelectrical properties of epoxy nano-silica composite sheets with an average thickness of 3 mm. Discussing the effect of epoxy reinforced with different nano-silica concentrations of 0, 5, 10, and 15wt% under different gamma radiation dosages of 1, 3, and 5 KGy. These AC electrical properties marginally increased with increasing doses of gamma irradiation in this study. This research has some application value. But the study still has some minor problems.

1.  The conclusion of the article is too concise, so it is necessary to refine the qualitative conclusion by sections.

2.  Why the curve of Figure 1 changes almost the same?

3.  What is the reason for the greater variation between the curve of nano-silica concentrations of 15wt% and the other three concentrations in Figure 5(c)?

4.  Figures 2 and 3 clearly show that the dielectric loss and dielectric constant are the smallest under gamma radiation doses of 3KGy and nano-silica concentrations of 15wt%. Why?

Author Response

Dear reviewer,

   Thank anonymous very much for your helpful comments on manuscript entitled “Effects of Gamma Radiation doses on the AC Electrical proper-ties of epoxy reinforced with nano-silica composites “to “journal of composites sciences”.

  • The changes of the first revision of manuscript are listed below.
  • Every change is highlighted by italic and green in the first revision of manuscript.

  • Author's Reply to the Review Report (Reviewer 1)

  1. The conclusion of the article is too concise, so it is necessary to refine the qualitative conclusion by sections.

   Modified as required, please read in conclusion section: (Epoxy nano-silica composites were irradiated with dosages of 1, 3, and 5 KGy by a Co-60 gamma ray source with a dose rate of 205.965 Gy/hr for a sample capacity of 4400 ml. The ac electrical properties of the composites were measured before and after irradiation. The results showed that the epoxy nano-silica composites were affected by the dosage of gamma radiation. The impedance values decreased for the gamma-irradiated composites. We found that gamma radiation can be used as a curing technique to decrease the electrical impedance of samples. The dielectric constant and dielectric loss of the composites also decreased after irradiation.

The gamma radiation process is a non-contact, non-thermal process that can be used to modify the properties of materials. Gamma radiation can be used to cure polymers, to create cross-links in polymers, and to induce an increase in ac electrical conductivity of the composites.

The gamma radiation process is a safe and environmentally friendly process that can be used to produce high-quality materials.)

  1. Why the curve of Figure 1 changes almost the same?

The behavior of the impedance for all the tested samples is the same in these figures but the values are changes.

  1. What is the reason for the greater variation between the curve of nano-silica concentrations of 15wt% and the other three concentrations in Figure 5(c)?

In figure 5, the variation between the curve of nanosilica of 15 wt.% are consistent, but the greater variation between the epoxy resin and other samples in this figure is obvious because the nano-silica content increase the ac conductivity

  1. Figures 2 and 3 clearly show that the dielectric loss and dielectric constant are the          smallest under gamma radiation doses of 3kGy and nano-silica concentrations of 15wt%. Why?

    Because Free radicals are atoms or molecules that have unpaired electrons. These unpaired electrons can interact with the electric field of the material, which can reduce the ability of the material to store electric charge. The decrease in the ability to store electric charge leads to a decrease in the dielectric constant. The effect of gamma radiation on the dielectric constant of epoxy composites is more pronounced at higher nano-silica concentrations. Please read lines (213-220) in the first revision of manuscript.

Reviewer 2 Report

Dear Authors,
I found your manuscript of high potential interest to readers due to the innovative technique of composites modification by gamma radiation.
Below, please find several remarks, which should be applied, in my opinion, to the manuscript, before publication:
- line 3 - some lack of spaces in the title words
- line 79 - should be "epoxy resin filled" instead of "epoxy filled"
- line 88 - correct "inthe"
- line 94 - correct "increasedwith"
- lines 100 and 102 - lack of spaces in "et al."
- lines 115-123 - I suggest moving this description to the Introduction section
- line 134 - I suggest saying "was incorporated to improve the AC" to "was incorporated to investigate the influence of the AC"
- figure 1 (a-d) - Please provide the same Y-axis scale to make it easier to compare the results. Same for the following plots
- to Conclusion section: please provide any practical implications of your findings, as well as any potential scenarios of continuation of your research

Best regards!

Author Response

Dear reviewer,  

   Thank anonymous very much for your helpful comments on manuscript entitled “Effects of Gamma Radiation doses on the AC Electrical proper-ties of epoxy reinforced with nano-silica composites “to “journal of composites sciences”.

  • The changes of the first revision of manuscript are listed below.
  • Every change is highlighted by italic and red in the first revision of manuscript.

Author's Reply to the Review Report (Reviewer # 2)

  • - line 3 - some lack of spaces in the title words : ( done )
    - line 79 - should be "epoxy resin filled" instead of "epoxy filled" : ( done )
    - line 88 - correct "inthe" : ( done )
    - line 94 - correct "increasedwith" : ( done )
    - lines 100 and 102 - lack of spaces in "et al.": ( done )
    - lines 115-123 - I suggest moving this description to the Introduction section: ( done )
    - line 134 - I suggest saying "was incorporated to improve the AC" to "was incorporated to investigate the influence of the AC" : ( done )

2) - figure 1 (a-d) - Please provide the same Y-axis scale to make it easier to compare the results. Same for the following plots

If the Y-axis is drawn with the same scale for all graphs, it can be difficult to distinguish between the values for each graph because the readings will be too close together. Therefore, I used a different scale for each graph to make it easier to see the values.

3) - to Conclusion section: please provide any practical implications of your findings, as well as any potential scenarios of continuation of your research

Modified as required, please read in conclusion section: (Epoxy nano-silica composites were irradiated with dosages of 1, 3, and 5 KGy by a Co-60 gamma ray source with a dose rate of 205.965 Gy/hr for a sample capacity of 4400 ml. The ac electrical properties of the composites were measured before and after irradiation. The results showed that the epoxy nano-silica composites were affected by the dosage of gamma radiation. The impedance values decreased for the gamma-irradiated composites. We found that gamma radiation can be used as a curing technique to decrease the electrical impedance of samples. The dielectric constant and dielectric loss of the composites also decreased after irradiation.

The gamma radiation process is a non-contact, non-thermal process that can be used to modify the properties of materials. Gamma radiation can be used to cure polymers, to create cross-links in polymers, and to induce an increase in ac electrical conductivity of the composites.

The gamma radiation process is a safe and environmentally friendly process that can be used to produce high-quality materials.)

Reviewer 3 Report

(1) The material proposed in this manuscript has unique AC electrical properties. What are the application scenarios for this material? It is recommended that the authors add some introductions in the Introduction section.

(2) What experimental instruments were used by the authors, and what are the specific operation procedures? Please clearly illustrate this using images or flow charts.

(3) The annotations in Figure 3 are unclear. Please, authors, self-check for similar issues.

(4) The title of Figure 5 has been placed on two different pages from the image. Authors are encouraged to self-check for similar formatting issues.

(5) The analysis and explanation provided in the text are overly brief. What are the specific advantages of the material proposed in this paper compared to other similar materials? The authors are asked to reflect on this.

(6) The Conclusion section of the paper is too concise. Could the authors elaborate on the future development directions of the proposed material and provide a more in-depth analysis of the innovative points in this paper?

Author Response

Dear reviewer,

   Thank anonymous very much for your helpful comments on manuscript entitled “Effects of Gamma Radiation doses on the AC Electrical proper-ties of epoxy reinforced with nano-silica composites “to “journal of composites sciences”.

  • The changes of the first revision of manuscript are listed below.
  • Every change is highlighted by italic and blue in the first revision of manuscript.

Author's Reply to the Review Report (Reviewer # 3)

  • The material proposed in this manuscript has unique AC electrical properties. What are the application scenarios for this material? It is recommended that the authors add some introductions in the Introduction section.

Modified as required, please read in introduction section : (The use of gamma radiation to alter the material properties of epoxy composites opens up a range of new possibilities. This technique has the potential to create advanced materials with improved electrical insulation, as well as improved electrical conductivity. The applications these materials could be used in range from high-voltage cabling and transformers to EMI shielding and heat sinks. By making tweaks to the properties of epoxy composites with gamma radiation, it could be possible to substitute the use of metals in heat transfer and the mitigation of electromagnetic interference. The potential applications for this technology are vast and could have far-reaching implications for materials development.)

  • What experimental instruments were used by the authors, and what are the specific operation procedures? Please clearly illustrate this using images or flow charts.

Modified as required, please read in samples preparation section:( To study the electrical properties of epoxy nano-silica composite, impedance (Z) and phase shift angle () measurements were performed using a Hewlett Packard (HP) 4192A Impedance Analyzer. The HP 4192A Impedance Analyzer measures impedance and phase angle as a function of applied frequency.)

  • The annotations in Figure 3 are unclear. Please, authors, self-check for similar issues.

Modified as required, please read in Results and discussion section lines ( 260-269)  : (The increase in the dielectric loss value, due to the gamma irradiation, is likely due to the increase in the number of dipoles created in the composites. This is because gamma radiation causes an increase in free electrons and electron-hole pairs, leading to more charges which then form dipoles. The more dipoles that are present in a material, the more energy will be lost when an electric field is applied. The increase in dielectric loss can have a number of effects on the material. For example, it can reduce the material's ability to store energy, and it can also increase the material's susceptibility to electrical breakdown. The effects of dielectric loss will depend on the specific material and the conditions under which it is operating)

  • The title of Figure 5 has been placed on two different pages from the image. Authors are encouraged to self-check for similar formatting issues.

- done

      (5) The analysis and explanation provided in the text are overly brief. What are the        specific advantages of the material proposed in this paper compared to other similar    materials? The authors are asked to reflect on this.

     Previous studies did not investigate the compound of epoxy and silica. Instead, they extracted the properties of silica and epoxy and studied how they interact electrically. Through research, it was concluded that silica particles are effective in improving electrical properties, especially under the influence of radiation. This is what we also found in our results.

      (6) The Conclusion section of the paper is too concise. Could the authors elaborate    on the future development directions of the proposed material and provide a more in-depth analysis of the innovative points in this paper?

Modified as required, please read in conclusion section: Epoxy nano-silica composites were irradiated with dosages of 1, 3, and 5 kGy by a Co-60 gamma ray source with a dose rate of 205.965 Gy/hr for a sample capacity of 4400 ml. The ac electrical properties of the composites were measured before and after irradiation. The results showed that the epoxy nano-silica composites were affected by the dosage of gamma radiation. The impedance values decreased for the gamma-irradiated composites. We found that gamma radiation can be used as a curing technique to decrease the electrical impedance of samples. The dielectric constant and dielectric loss of the composites also decreased after irradiation.

The gamma radiation process is a non-contact, non-thermal process that can be used to modify the properties of materials. Gamma radiation can be used to cure polymers, to create cross-links in polymers, and to induce an increase in ac electrical conductivity of the composites.

The gamma radiation process is a safe and environmentally friendly process that can be used to produce high-quality materials.

Round 2

Reviewer 1 Report

The author made modifications according to my opinion, and I agree to be accept.

Reviewer 2 Report

Dear Authors,
thank you for providing reasonable response to the remarks mentioned in the review, by adding sufficient changes in the manuscript.
No other remarks from my side.
Best regards!

Reviewer 3 Report

No comment